# Mothers' and Grandmothers' misconceptions and socio-cultural factors as barriers to exclusive breastfeeding: A qualitative study involving Health Workers in two rural districts of Ghana

**Christiana Nsiah-Asamoah**[1]*, **David Teye Doku**[2], **Samuel Agblorti**[2]

1 Department of Clinical Nutrition and Dietetics, University of Cape Coast, Cape Coast, Ghana,
2 Department of Population and Health, University of Cape Coast, Cape Coast, Ghana

* cbuxton@ucc.edu.gh

## Abstract

### Background

Education on exclusive breastfeeding (EBF) practices is usually given in the form of health talks by health workers (HWs). The need for HWs to be well-informed about cultural practices and misconceptions that act as barriers to EBF has been documented in literature. This information can guide HWs in developing interventions such as health talks which are culturally sensitive. However, this has not been explored from the perspectives of HWs in Ghana. In this paper, we report mothers' and grandmothers' misconceptions and cultural practices that are barriers to EBF in two rural districts in Ghana from the perspectives of Community Health Workers and Community Health Volunteers.

### Methods

We used qualitative data collected in the Kwahu Afram Plains South and North Districts of Ghana through nine focus group discussions (FGDs) among HWs and followed the data saturation principle. All FGDs were audio-taped, transcribed verbatim and translated from local dialects to English. The emerging themes were used in writing a narrative account, guided by the principles of the thematic analysis.

### Results

Our main findings included mothers' and grandmothers' perceptions that HWs themselves do not practice EBF. Mothers had the perception that grandmothers did not practice EBF but their children grew well, and gestures of babies suggested their readiness to start eating. Misconceptions revealed included beliefs that breastmilk is watery in nature and does not satisfy infants. Another misconception was that babies gain weight faster when not exclusively breastfed but fed on infant formulas. A custom of giving corn flour mixed with water or light porridge during the first few days after birth to welcome newborns was also reported.

**Data Availability Statement:** The recorded audiotapes that were translated and transcribed

into English from the focused group discussions cannot be shared publicly because it contains sensitive and some personal identifying information from the participants. However, data are available from the District Health Management Committee on Human Research and Ethics (Cordinator's Email: davekwame066@gmail.com) or the corresponding author (Email:cbuxton@ucc.edu.gh) upon request from researchers who meet the criteria for access to confidential data.

**Funding:** The authors received no specific funding for this work.

**Competing interests:** The authors received no specific funding for this work.

**Abbreviations:** CHW, Community health worker; CHV, Community health Volunteer; CWCs, Child Welfare Clinics; EBF, Exclusive Breastfeeding; FGD, Focus group discussion; GDHS, Ghana Demographic and Health Survey; GHS, Ghana Health Service; HWs, Health Workers; KAPND, Kwahu Afram Plains North District; KAPSD, Kwahu Afram Plains South District; UNICEF, United Nations Children's Fund; WHO, World Health Organization.

## Conclusions

The reports of the HWs revealed that several socio-cultural factors and misconceptions of mothers and grandmothers negatively influence EBF practices of mothers. Findings from this study highlight the need for HWs to provide culturally appropriate counselling services on breastfeeding not only to mothers but also to grandmothers and fathers in order to promote EBF and reap its benefits.

## Introduction

According to the United Nations Children's Fund (UNICEF), with respect to breastfeeding, every second counts in an infant's life. Hence, the time at which it is initiated and its exclusivity for the first six months of life can make a whole lot of difference between life and death [1]. Early initiation of breastfeeding, that is, putting newborns to the breast within the first hour of their lives safeguards their survival and considerably reduces the risk of death during the critical neonatal period of life [2]. In addition, early initiation of breastfeeding has been described as an effective intervention that helps to establish exclusive breastfeeding (EBF) [3]. Exclusive breastfeeding for the first six months of life is a recommended intervention, in view of its established benefits of reducing the risks of morbidity and mortality in infants [3]. There is also evidence that EBF protects against pneumonia and diarrhoea–the two leading killers of children under five [1].

While breastfeeding is common in most parts of the world, EBF is not the norm. The situation is not different in Ghana where, according to the latest Ghana Demographic and Health Survey (GDHS) report of 2014, almost all children in Ghana (98 percent) are breastfed at some point in their life [4]. However, only 52 percent of children less than 6 months of age are exclusively breastfed; and the median duration of EBF is approximately four months instead of the recommended six months. A cursory study of previous GDHS reports reveals that, with regard to EBF, the prevalence increased considerably from 53% in 2003 to 63% in 2008, but declined to 52% in 2014 [4–6]. Therefore, according to the 2014 GDHS, the percentage of children aged between 0 and 5 months who were exclusively breastfed has decreased by approximately 17% between 2008 and 2014. It is important to acknowledge that the findings in the 2014 GDHS, which was the only available information representative of Ghana at the time of the study (in 2018), may not be an updated information. However, these figures suggest that the practice of EBF might be gradually declining among mothers and requires much effort to assess the factors leading to this occurrence. In addition, the prevalence of EBF reported in these demographic and health surveys obviously falls far below the widely accepted "universal coverage" target of 90% coverage [7].

The barriers which have been identified as preventing EBF are generally maternal, child and socially or environmentally-related. Maternal-related barriers to EBF include low educational level of mothers, working mothers, difficulties in obtaining maternity leave, increase in mother's workload, less number of antenatal visits and lack of knowledge about EBF [8]. Child-related factors such as being a male and having been delivered by cesarean section have been reported to reduce the likelihood of being exclusively breastfed [9]. Social and environmental barriers to EBF which include lack of spousal support with chores, controversial health messages on EBF delivered at health facilities and unsupported environments for EBF in public places have been identified as discouraging EBF [10, 11].

It has been indicated that most infants are not exclusively breastfed but are given other foods or liquids throughout the first six months, due to cultural practices and beliefs and some misconceptions held by mothers [12]. These cultural practices, myths, misconceptions and beliefs influence the ability of most mothers to initiate breastfeeding within one hour after birth, practice EBF and prolong the duration of breastfeeding, particularly in low-income countries [13]. In Ghana, Tampah-Naah and Kumi-Kyereme [14] assert that the limited practice of EBF in all regions of the country can be attributed to cultural beliefs. However, there is little information regarding these cultural practices and misconceptions which are barriers to the practice of exclusive breastfeeding for the first six months in a developing country like Ghana. Most studies undertaken in the past to investigate cultural practices influencing optimal EBF practices were conducted among mothers [12–16] and grandmothers [16, 17]. For example, in the study of Thet et al. [16] which was undertaken in the Ayeyarwaddy Region of Myanmar (formerly Burma), the barriers to EBF were assessed qualitatively from the views of mothers, grandmothers, and husbands. Clearly, there has been less focus on context, norms and cultural practices influencing EBF from the viewpoints of health workers (HWs). However, recommendations from studies suggest the need for health professionals in various settings to understand and be well-informed about these cultural practices and misconceptions that act as barriers to appropriate breastfeeding practices [18, 19].

In Ghana, there is a dearth of information on barriers to EBF practices of mothers from the perspectives of Community Health Workers (CHWs) and Community Health Volunteers (CHVs) who play a major role in providing care to breastfeeding women, neonates and young infants when they undertake both home visits and community outreach programmes. We seek to fill this gap by assessing the perceptions of these group of HWs regarding socio-cultural influences and mothers' misconceptions about EBF. Although, other key health professionals like midwives and nurses play significant roles in promoting child health and nutrition, the study focused on only CHWs and CHVs because of some reasons. The study concentrated on only CHWs and CHVs because they play an important role in the delivery of health care services particularly in rural settings where there are shortages of trained health care workers. Both CHWs and CHVs have been described as agents of behaviour change promotion as they interact with community members in their allocated catchment areas [20]. They are regarded as being in a better position to deliver crucial health messages, empower community members to enable them make informed decisions and therefore increase local access to health preventive measures [21].

Therefore, a better understanding of cultural practices and misconceptions that negatively imparts EBF practices of mothers can particularly guide CHWs and CHVs in developing interventions such as health talks which are culturally sensitive to help disabuse the minds of caregivers on issues regarding EBF. In addition, the findings can play a role in the development and implementation of some intervention activities by the HWs on the basis of the Motivational theory of role modeling [22] which highlights the power of role models (in this study, HWs) to serve as sources of inspiration and agents of behavioural change to their clients (in this study, lactating mothers). Therefore, this study assessed the cultural practices and misconceptions of mothers that influence EBF practices from reports of CHWs and CHVs (collectively referred to as HWs for the purpose of this study) working in two rural districts in Ghana.

## Subjects and methods

### Ethics

Ethical approval for this study was granted by the Dodowa Health Research Centre (DHRC) Institutional Review Board (IRB) of the Ghana Health Service (Reference/Identification: DHRCIRB/04/02/17)and the Institutional Review Board (IRB) of the University of Cape Coast

(U.C.C) (Reference/Identification: UCCIRB/CHLS/2017/02). Participants who gave their consent to participate in the FGDs willingly signed a consent form.

## Study area

The study was conducted in two rural districts—Kwahu Afram Plains North (KAPND) and Kwahu Afram Plains South (KAPSD) in the Eastern Region of Ghana. Kwahu Afram Plains North District (KAPND) is located in the northern-most part of the Eastern Region and the capital of the district is Donkorkrom. With the exception of Donkorkrom, which is urban, the rest of the settlements in the district are peri-urban, rural and small communities. The district has 86% rural and 14% urban population distribution. Approximately, three quarters of the communities are located on Islands within water bodies (Volta lake, River Afram and Obosom River). Mostly, the economy of KAPND is an agrarian one. Within the employed labour force, the major occupation in the district is agriculture which is largely subsistence in nature (employing 74.5% of the total labour force). Regarding health services infrastructure, the district has one hospital at Donkorkrom and thirteen Community-based Health Planning Services (CHPS). According to the Kwahu Afram Plains North District Planning Coordinating Unit (DPCU) 2018 report, most of the inhabitants in other settlement communities take between 25 and 35 minutes of walking to access the main district hospital at Donkorkrom. About 59.9% of the population patronize health facilities outside the settlement they live [23].

Kwahu Afram Plains South District (KAPSD) is located at the north-western part of the Eastern Region and Tease is the administrative capital town of the district. About 85.3% of the population aged 15 years and older are economically active. Of the employed population, majority (80.0%) are engaged as skilled agricultural, forestry and fishery workers.

With respect to health services infrastructure, the KAPSD has six (6) health centers, 15 Community-based Health Planning Service (CHPS) zones and one (1) private clinic. The only referral health center in the district is the Presbyterian Health Center at Tease. Similar to KAPND most of the inhabitants have to walk between 25 and 35 minutes to access the main health center at Tease [24]. Ofosu's [25] study in the Eastern Region revealed that several localities in the Afram Plains districts are beyond the 5 kilometres recommended distance by the Ghana Health Service which is considered as good for people to access health care from a health facility.

## Study design and population

The study was descriptive and cross-sectional in which qualitative data was collected through nine (9) Focus Group Discussions (FGD) in two selected districts in the Eastern Region of Ghana: Kwahu Afram Plains North (KAPND) and Kwahu Afram Plains South (KAPSD). These two districts were purposively selected from the 26 districts in the Eastern Region on the basis of being ranked respectively as the first and second districts with the highest prevalence of underweight among children under five (5) years in the Region in both 2015 and 2016.

The CHWs who participated in the FGDs were selected from 21 randomly sampled Child Welfare Clinics (CWCs) which are found in health facilities within the districts. The CHVs were purposively selected with the assistance of the District Nutrition Officers who knew CHVs who had worked for more than 5 years and who were actively involved in promoting child nutrition health services within the districts. In this study, the HW should have been working for at least five years in the district before he or she participates in the FGDs because of the assumption that these number of years might have given them ample experiences to be able to provide information on socio-cultural issues influencing EBF within the selected districts. A total of 78 health workers (HWs) were recruited face- to- face with assistance from the district nutrition officers and nurse-in-charges in the various CWCs to participate in the

FGDs. Forty-two (42) CHWs were selected from 21 CWCs and 36 CHVs were chosen from 21 communities within the districts.

## Data collection

The data was collected in the districts between 6<sup>th</sup> September and 15<sup>th</sup> December, 2018.

Translation and transcription of the data were done between January and May 2019.

All the nine sessions of the FGDs were held at the premises of the District Health Directorate on different days.

The Focus Group Discussion guide was a modified version from previous related studies [13, 26, 27], which were also undertaken to access cultural factors that influence infant feeding practices. The FGD guide was pretested to assess for reliability, clarity and simplicity of the tool among six health workers who were recruited from two health centres. The FGD guide comprised of six(6) discussion topics. The discussion questions included HWs opinions on socio-cultural factors that prevent early initiation of breastfeeding within at least 30 minutes after birth and exclusive breastfeeding of infants for the first six (6) months of life. Other discussion questions focused on HWs' opinions on social influences exerted by significant persons in households that prevented mothers from adopting optimal breastfeeding practices, foods commonly introduced to infants before they attain six (6) months of age and why they are given to babies. The FGD guide had a question that gave the HWs an opportunity to share their observations and experiences related to feeding of infants below two (2) years during their home visits and community health outreach activities.

A total of nine (9) FGDs were conducted among the CHWs and CHVs in groups of between 8 and 10 members. The FGDs were conducted in the predominant languages of the people—Ewe, Twi and Frafra. Each FGD took approximately 1 hour 10 minutes to complete and they were conducted until the point of data saturation. After completing the eighth FGD, the research team had a discussion and realized that data saturation had been achieved since no new information was emerging from the study participants and therefore data collection was stopped after the nineth FGD. The proceedings of all FGDs were audio-recorded; and detailed notes and pictures were taken. The first author and a Public Health Nurse acted as the facilitators, with the help of three trained research assistants who were technical officers working in the district health directorate. Two of the research assistants took notes and ensured that the session was audio-recorded, whiles one helped with translation of the questions into the local dialects when it was necessary.

## Analysis of data

Qualitative thematic analysis techniques as described by Braun and Clark [28] were employed in the data analysis. The audio records which were in local dialects (Twi, Ewe and Frafra) were translated and transcribed into English. The first step of the thematic analysis involved familiarization with the qualitative data. The transcripts, with the hand-written notes taken during the FGD sessions, were read several times to make better meaning out of the responses and were compared across the different FGDs by two independent translators (Research Assistants). The second step after familiarization of data entailed coding which was done independently by the authors and subsequently compared. All discrepancies were resolved by the authors and the codes were refined until no new codes were identified. The coding involved identifying key words and phrases that best described and clarified each response from the FGDs. The third step involved the generation of themes. A matrix table was used to list the codes generated and all related codes were categorised under one theme. The themes were then reviewed and refined to ensure that they (themes) encompassed all the codes. The refined

themes were eventually reported narratively and discussed. It was also ensured that the various statements and interpretations given under each theme reflected the implications gotten from the responses of the HWs. NVivo 10 qualitative software was used for the analysis.

## Research team and reflexivity

The two moderators of the FGDs had a background in Public Health Nutrition and Public Health Nursing. The two field assistants who took field notes and did the translations had a background in Disease Control and Health Informatics. It is important to note that the lead researchers and the team were all Ghanaians, but not indigenous people from the study area. The diverse background of researchers was to ensure reflexivity. There was also a rapport between the researchers and study participants and as a result of that, study participants were more willing to share their perceptions and experiences. Additionally, coding and interpretation of data were also discussed among researchers who had backgrounds in Nutrition, Social Science and Public Health. This eventually reduced personal biases of the researchers and therefore reduced influences of the researchers on the data outcomes.

## Results

The demographic information of the HWs who participated in the study is presented in Table 1.

**Table 1. Demographic information of study participants.**

| Characteristics | Frequency (%) |
|---|---|
| **Sex** | |
| Female | 51 (65.4) |
| Male | 27 (34.6) |
| **Category of Health Worker** | |
| Community health Workers | 42 (53.8) |
| Community health Volunteers | 36 (46.2) |
| **Work experience (in years)** | |
| 5–10 | 24 (30.8) |
| 11–15 | 38 (48.7) |
| >15 | 16 (20.5) |
| **Participation in any child nutrition training workshop (n =??)** | |
| Yes | 67 (85.9) |
| No | 11 (14.1) |
| **Number of times participated in child nutrition workshop (n = 67)** | |
| Once | 21 (31.3) |
| Twice | 33 (49.3) |
| Thrice | 13 (19.4) |
| **Theme focus of child nutrition workshop attended** | |
| Breastfeeding counselling and lactation management | 46 (36.5) |
| Community management of Acute Malnutrition (CMAM) | 41 (32.5) |
| Complementary Feeding | 39 (31.0) |

The results presented in Table 1 show that majority of the study participants were females (65.4%) and had worked between 11 and 15 years (48.7%). Again, a higher (85.9%) proportion had participated in a child nutrition workshop. Out of those (67) who had participated in a child nutrition workshop, almost half 33(49.3%) had attended such workshops on two occasions. These workshops were focused on breastfeeding counseling, Community Management of Acute Malnutrition (CMAM) and on complementary feeding.

The findings that emerged from the FGDs are presented under three main themes: perceptions of mothers that prevent them from practicing EBF, misconceptions of mothers with regard to EBF and cultural practices in the community that prevent mothers from exclusively breastfeeding their babies. Most of the participants reported that they learnt about these misconceptions and beliefs from mothers they had come into contact with at their Child Welfare Clinics (CWCs). It is important to note that all the statements and reports presented in this study are perceptions of the HWs based on their interactions with mothers, grandmothers and observations made during their outreach programmes and home visits within their communities of operation.

## Perceptions of mothers that prevents them from practicing EBF

Respondents were asked to deliberate over perceptions of mothers that prevented them from practicing EBF. The items that guided the discussion programmes solicited for information on some perceptions held on by mothers that prevented them from exclusively breastfeeding their babies for the first six months of their lives. According to the HWs some of the perceptions of mothers were that they were of the view that HWs themselves do not practice EBF and that their grandmothers and great grandmothers did not practice EBF. The statements of some of the HWs are captured below.

**HWs themselves do not practice EBF.** The following statements by CHVs from different communities demonstrate that some mothers have the perception that health workers (HWs) themselves do not exclusively breastfeed their children, but they advise other mothers to practice EBF.

One CHV reported: "*I once encountered a mother who told me that she was not practicing EBF because even nurses who told mothers to practice EBF were not practicing it themselves*". (CHV 2, FGD 2 KAPND).

Another CHV reported: " *During one of my home visits, I came across a grandmother giving maize porridge to a baby who was about four weeks old. When I asked the mother why she was not giving only breastmilk, she replied that the nurses do not want to tell them the truth about feeding babies under 6 months, because they think mothers do not take good care of their feeding bottles. As a result they advise mothers to breastfeed their babies until 6 months. But she has seen nurses who give lactogen to their babies when they are less than 6 months"* (CHV 1, FGD 3 KAPND).

**Grandmothers and great grandmothers did not practice EBF.** According to the CHWs and CHVs, another barrier that prevented mothers from exclusively breastfeeding their babies was the perception of some grandmothers that they never practice EBF on their own children but they grew well as expected; and they were also healthy.

The following reports were made by some CHWs and CHVs:

One CHV narrated some encounters she had with grandmothers during her home visits activities. She stated: "*During home visits, grandmothers would always complain that they fed their own babies under 6 months with porridge and soup, and nothing happened to them. So why are health workers worrying them with EBF*?"(CHV 1, FGD 2 KAPSD).

The CHWs/CHVs indicated that some mothers continue to tell them that they have been feeding their babies on light food before they attain 6 months of age, and nothing happens to them. So why should they practice EBF?

One CHW indicated: "*Some mothers with babies under 6 months whom they feed with various food items would tell you that this was what our mothers and grandmothers did, and there was no problem with their children. So if they do the same, there would not be any problem*". (CHW 2, FGD 4 KAPND).

## Misconceptions of mothers reported by health workers

The HWs were asked what they perceived as some misconceptions held by mothers that discouraged them from practicing EBF. The findings indicate that some of the misconceptions held by mothers with respect to EBF are that babies cannot survive without water and therefore must be given water in addition to breastmilk. Another misconception of some mothers is that breastmilk is watery in nature and does not contain enough food to satisfy infants. Again mothers thought gestures and actions of babies suggest their readiness to start eating food. In addition, mothers had the misconception that babies who are not exclusively breastfed gain weight better and faster than those who are exclusively breastfed. According to the HWs, some mothers were of the view that breastmilk does not contain enough water to satisfy the thirst of babies. So, they give them water after feeding them just as adults usually drink water after eating. The responses of the CHWs and CHVs on these misconceptions of mothers are reflected in the following statements:

**Babies cannot survive without water.** According to the HWs, some mothers had the perception that water is a basic necessity in life and therefore it is impossible for a baby to survive for the first six months of life without drinking water. The following reports by the HWs demonstrate that some mothers do not accept and believe that a newborn baby can survive solely on breastmilk to meet their water requirements for the first six months of life.

One CHW, narrated an interaction she had with a mother who had a 2-month-old baby, on the need to practice exclusive breastfeeding. According to the CHW, "*this mother openly refused to exclusively breastfeed her baby, because she lost a child when she practiced it for four months without giving the baby any water. She thinks the child died as a result of exclusive breastfeeding. Therefore, she is not encouraged to practice it again on any of the children she would have*". (CHW 2, FGD 4 KAPND)

Another CHV indicated that: "*Some mothers always say their mothers, grandmothers and great grandmothers gave water to their babies, and nothing happened to their children. So there is no way they are going to change this trend of giving water to their babies*".(CHV 2, FGD 4 KAPSD)

**Breastmilk is watery in nature and does not contain enough food to satisfy infants.** According to HWs, some mothers who do not exclusively breastfeed their babies suggest that breastmilk is watery in nature and does not contain sufficient food to satisfy infants.

One CHW stated: "*Some mothers give other foods such as porridge mixed with groundnut paste or fish powder to infants less than 6 months, because they believe breastmilk is watery and does not satisfy the hunger needs of a child*". (CHW 1, FGD 4 KAPSD)

Another CHW reported: "*Some mothers also claim that there is a great difference between breastmilk and water. Yet health workers continue to say that breastmilk contains adequate water for a baby. Hence there is no need to give water to children below six months of age. Most mothers are skeptical of this assertion about the water content of breastmilk*". (CHW 2, FGD 4 KAPND)

According to the HWs some mothers also give water to infants less than 6 months, because they claim the hot weather makes babies thirsty and uncomfortable, exhibited in frequent crying and inability to sleep soundly.

The reports of some of the HWs are presented in the statement below:

**Mothers' belief that the hot weather makes babies thirsty.** According to one CHW: "*Some mothers express fears that children will become dehydrated if they are not given water to drink, considering the hot weather condition that makes one thirsty often*" (CHW 2, FGD 5 KAPND)

**Male babies may not be satisfied when they solely feed on breastmilk.** It was also revealed from the responses of the HWs that some mothers with male babies had the

misconception that they may not be satisfied when they are only fed on breastmilk as compared with female babies.

One of the CHW's reported that: " *Most mothers with male babies either exclusively breastfeed for just about one month or never practice exclusive breastfeeding on their male babies. They claim boys breastfeed for longer periods; and they doubt whether male babies are satisfied by only depending on breastmilk for the first six months of their lives*". (CHW 2, FGD 2 KAPND)

**Gestures and actions of babies suggest their readiness to start eating food.** Some of the HWs indicated that some mothers say that whenever they eat the child looks at them intently as if they also want to eat. So, they are compelled to give some of the food to the child.

One CHW narrated: "*Some mothers say their babies put their hands or other objects into their mouth and chew them, while others say their babies chew and suck their fingers; and this suggests that the child is ready to eat other foods and not only breastmilk.* " (CHW 1, FGD 5 KAPND)

Another CHV stated: " *Some mothers also say that there are situations when babies cry a lot when they see them or other family members eat and this suggests that they want to start eating. So, they give them some of the food to eat*" (CHV 2, FGD 3 KAPSD)

**Babies gain weight faster when not exclusively breastfed but when fed on infant milk formulas.** The responses of the CHWs and CHVs on this misconception of mothers are reflected in the following statements:

One of the CHV stated: "*there are situations in which a mother may not be practicing EBF and her child gains weight and grows better than children of mothers practicing EBF. This discourages other mothers from practicing EBF."*(CHV 2, FGD 5 KAPND)

Another CHW indicated that "*There are also cases where mothers who have money, decide to feed their babies on "Lactogen" (an infant milk formula), because they believe it enables infants to grow faster, bigger and chubbier than when solely fed on breastmilk*".(CHW 2, FGD 4 KAPND)

## Cultural practices that prevent EBF

The HWs were asked to share some cultural practices that they perceived as discouraging mothers from practicing EBF, on the basis of their observations during home visits and community outreach programmes. Most of the CHWs and CHVs indicated that, generally, mothers do not exclusively breastfeed their babies for the first 6 months of their lives. Some mothers practice EBF for only a month or two and stop for a number of reasons. The findings indicate that some rituals that are performed before babies are allowed to come out in public and giving certain foods to babies during the first few days after birth to welcome them prevented mothers from exclusively breastfeeding their babies. The statements presented below indicate some of the reasons reported by the CHWs and CHVs for the break in the practice of EBF.

**Performing rituals for newborns before they are allowed to come out in public.** The following narrations by some CHVs indicate that some rituals performed for newborns before they can be seen in public may also prevent EBF:

One CHW asserted: "*In some idol worshipping homes, every newborn child owes allegiance to the gods. As a result, certain rituals must be performed by giving a newborn baby some herbal concoctions during the first week after birth which are believed to protect, cleanse the internal body system and give some extraordinary powers to the child to scare away evil spirits before it is allowed to come out in public or show up at CWCs*". (CHW 2, FGD 5 KAPND).

Another CHV further commented on this practice of hiding newborns until rituals are performed for them: *"This practice is common among mothers who deliver at home and not in a health facility".* (CHV 2, FGD 3 KAPSD)

**Giving certain foods to babies during the first few days immediately after birth to welcome them.** According to the CHWs and CHVs who participated in the FGDs, there are some cultural practices and customs that lead to the giving of certain foods to babies during the first few days immediately after birth to welcome them. According to the HWs, these customs prevented mothers from exclusively breastfeeding their babies. As one CHV said: "*In some communities, as a tradition and by custom, when a child is born, corn flour is mixed with water and given to the child to welcome him or her with the following statement*: *This is what we have been eating before you arrived, so if you have joined us today you are also going to eat the same.*" (CHV 2, FGD 2 KAPND).

Another CHW narrated: "*There are some cultural practices which demand that light porridge is given to newborn babies during the first few days after birth to welcome them, because it is believed that the child has travelled over a long distance into this world and, for that matter, is hungry.*" (CHW 2, FGD 4 KAPSD)

Another CHV stated: "*For children under 6 months, some mothers also boil 'Jathrofa* (a flowering plant) ' *with 'negro pepper'(Xylopia aethiopica) in a pot and give it to babies. They believe that this enables the child to sleep well and protects the child against diseases. They give negro pepper* (*Xylopia aethiopica*), *locally called 'hwentia', because they believe it heals sores in the stomach of newborns and prevents diarrhoea in babies*" (CHV 2, FGD 4 KAPND)

One CHV also reported: "*Some mothers give shea butter mixed with warm water to their babies who are less than 6 months to drink. They believe it enables them have free bowels.*" (CHV 3, FGD 2 KAPSD)

## Discussion

The perceptions and reports of Community Health Workers (CHWs) and Community Health Volunteers (CHVs) regarding the cultural practices and misconceptions of mothers that inhibit 6-months of EBF were explored through FGDs. According to the health workers, some misconceptions held on to by mothers prevented them from exclusively breastfeeding their babies.

The findings revealed HWs report that some mothers had the misconception that HWs themselves do not practice EBF. The statements from the HWs imply some mothers had the perception that nurses who advised them to exclusively breastfeed their babies were unfortunately going contrary to their own advice, which resulted in not encouraging some mothers to practice EBF. Although, the HWs did not confirm or refute this misconception and as to whether this perception is the real case, this report from the HWs has implications on the services that they provide to help mothers care for their babies. However, in support of the reports by mothers to this effect, studies that were conducted in Brazil [29] and Northwest Ethiopia [30] also found out that some health professionals were not adhering to the recommended period of exclusive breastfeeding.

In the study that was carried out in Northwest Ethiopia [30] which assessed EBF practices of nurses and midwives, it was found that though they had adequate knowledge on breastfeeding, their high knowledge level was not translated into their practice of EBF. The exclusive breastfeeding rate among the nurses was found to be 35.9%; and almost half (49.4%) of them exclusively breastfed their babies for only three months or less. Likewise, a study undertaken among health professionals in an accredited baby friendly hospital in Brazil revealed that only 28.3% sustained exclusive breastfeeding for the first six months of their babies' lives [29].

Some of the common reasons mentioned by the nurses and midwives for their failure to exclusively breastfeed their babies included work-related problems, for instance, short three (3)-months duration of maternity leave, lack of nearby child care facilities and inflexible work

schedules that prevented them from taking nursing breaks [29, 30]. It is important to note that these past studies [29, 30] were quantitative studies which were aimed at assessing the prevalence of EBF among health professionals and no information was obtained with respect to socio-cultural factors influencing EBF practices.

These reports however may suggest the need for HWs to act as "role-models" in order to encourage mothers to heed their advice as most mothers greatly rely on health workers' advice on infant feeding, and consult HWs whenever they experience breastfeeding challenges. The recommendation on the need for HWs to act as role-models is based on the Motivational theory of role modeling [22] which has implications on the findings of this study. Role models through their exemplary deeds are often described as motivating others to set and achieve ambitious goals to follow their examples. The Motivational Theory of Role Modeling, highlights ways in which the influence of role models can be employed to improve role aspirants' motivation, strengthen their existing goals, and facilitate their acceptance of new goals [22]. With regard to the applicability of the motivational theory to this study, the role modelling responsibility of nurses can be harnessed to increase the motivational level and facilitate the adoption of new practices (EBF) by role aspirants' (in this study, lactating mothers).

Another misconception is based on the perception of mothers that their grandmothers and great grandmothers did not practice EBF, but their children survived and grew as expected. The responses of the HWs suggest that mothers in these rural settings are likely to tap into the experiences of grandmothers who may influence the breastfeeding practices of their daughters when they bring their own infant feeding beliefs and practices to support their daughters to enable them care for newborns. In a related study that was undertaken in rural areas in Kenya, mothers indicated that their parents advised them to give sugary water and other pre-lacteal foods to their babies, and it was difficult to ignore their advice. This was because their parents had gone through it before. They did not exclusively breastfeed their children. Yet, they grew as expected and were healthy [31].

An interpretation that can be given to the finding that mothers are likely not to practice EBF as a result of advices that are given by grandmothers can be based on the theory of planned behaviour. The theory explains the relationship between the intention to exclusively breastfeed, and the factors that affect a women's final decision regarding exclusive breastfeeding. The theory explains how significant persons (in this case grandmothers) can determine and influence behavioural change and the adoption of new health behaviours. In this case, grandmothers might be perceived by mothers as good advisers, because of their knowledge of existing social norms, past experiences and this may influence their infant feeding choices. The implication of this finding emphasizes the need for HWs to also target grandmothers when planning and conducting health education programmes on child nutrition.

The responses of the HWs suggested that some mothers had wrong perceptions about the water content of breast milk perhaps because they do not have adequate knowledge about the water and nutrient composition of breastmilk. Similar to the reports of the health workers, other related studies reported that mothers had the perception that breastmilk alone was insufficient for their babies. Rather, mother believed that water in addition to breast milk was necessary to hydrate infants and to quench their thirst [32, 33]. The findings suggest that if HWs are able to address this knowledge gap of mothers using evidence to support their assertions; it may reassure mothers that the water and nutrient needs of their babies can be met by practicing EBF.

According to the HWs, some mothers also had the perception that gestures and actions by babies suggested their readiness to start eating other family foods. This finding is consistent with other studies that were carried out in Australia which explored why mothers introduced solid foods early [34, 35]. A major reason cited for the commencement of complementary feeding prior to six months of age was gestures and cues exhibited by infants, suggesting their

readiness to start eating food [34, 35]. In these studies, mothers explained that these signals indicated that infants "wanted" or "were ready" for solid foods, regardless of their age. These findings imply that mothers have to learn and understand infant's cues, gestures and behaviours regarding specific needs. This is especially important during the first few months of infancy when they are familiarizing themselves with a newborn baby. These reports by HWs suggest that the interpretations given to infant gestures and behaviours influence early introduction of foods and the inability of mothers to exclusively breastfeed their babies. This finding suggests the need to include in the health education of expectant mothers' and "first-time" mothers' interventions that focus on accurate interpretation of infant cues, gestures and behaviours as putative weaning signs.

There were reports that some mothers with male babies are unable to exclusively breastfeed them. Their argument is that since boys eat a lot as compared with girls, it is doubtful whether they are satisfied when fed solely on breastmilk for the first six months of their lives. In support of this assertion by some mothers, it was found out in some previous studies in Zimbabwe [36] and Denmark [37] that girls were introduced to complementary foods later than boys when they were less than six months old. In a related study conducted in Kenya, it was reported that mothers with male children complained that they breastfed often which sometimes made them feel dizzy after suckling. As a result, there was a greater propensity to an earlier introduction to other foods and shorter breastfeeding duration with boys than girls [13]. This misconception of some mothers that have male babies may be as a result of low or lack of knowledge about the potential capacity of breastmilk to optimally support growth and prevent malnutrition in infants regardless of the sex of the child [38]. Another reason that can be attributed to this misconception is perhaps the lack of guidance and encouragement given by HWs to mothers to enable them practice EBF. This misconception perhaps suggests the need for HWs to use scientific evidence to support the information shared with mothers. Alternatively, with the permission of mothers who have exclusively breastfed their male babies, HWs can use their children as examples during CWC sessions to encourage other mothers to also practice EBF.

This finding has implications for the need to intensify nutrition counseling services by HWs to boost the confidence of mothers that when they eat healthy foods, they will be in a better position to produce more nutritious breastmilk as reported in a systematic review paper by Bravi et al. [39] to satisfy their babies, regardless of their sex, while remaining healthy and strong as well. This misconception perhaps also suggests the need for HWs to use scientific evidence to support child feeding and nutrition information or recommendations that are shared with mothers. Alternatively, with the permission of mothers who have exclusively breastfed their male babies, HWs can use their children as examples during CWC sessions to encourage other mothers to also practice EBF.

The Health workers also reported the practice of giving new-born babies some herbal concoctions believed to confer protection and give some powers that fights against wicked spirits before they appear at public places such as at CWCs. The giving of herbal concoctions which is an example of pre-lacteal feeding have been shown to delay the initiation of breastfeeding [40], interfere with exclusive breastfeeding and makes it difficult for breastfeeding to be established in newborns. The use of herbal concoctions has been found to make newborns more prone to early risk of severe gastrointestinal infections [41] which can result in loss of appetite and consequently reduce their intake of breastmilk. These reports by the HWs have health and safety implications given that there are grave concerns with respect to the dosage and safety of herbal preparations. There is also the possibility that infants with their immature digestive system may have difficulty digesting, absorbing and utilizing these herbal preparations.

Another misconception of mothers reported by HWs is that babies who are not exclusively breastfed gain weight faster and grow better. This misconception is perhaps as a result of the

perception that breastmilk alone does not satisfy babies and therefore might not support the attainment of any substantial weight gain in infants. In support of these assertions by mothers, some studies have revealed that longer duration of exclusive breastfeeding is negatively related to infant weight gain [38, 42]. In a systematic review report [42], it was concluded that duration of more than 4 months of exclusive breastfeeding may be associated with a reduction in weight gain in late infancy.

An explanation that can be given for the rapid weight gain in children who are not exclusively breastfed for six months but are quickly introduced to infant formula and other foods is their higher protein intake [42]. Hence, it is assumed that exclusively breastfeeding for the recommended period of six months may result in slower weight gain in later infancy. These findings suggest that HWs must emphasize more on the future or long-term benefits of EBF, such as its being protective against overweight and obesity and Type 1 Diabetes especially during the adulthood stages of life [43]. Regarding Type 1 Diabetes, the human milk contains substances that promote maturation of the immune system, conferring protection against its onset later in life [43].

The reports by the HWs also suggest that certain traditions and rituals that are performed for newborns, sometimes before a mother can start breastfeeding, could delay early initiation of breastfeeding, prevent EBF and lead to prelacteal feeding practices. An example is the practice of mixing corn flour and water which is given to newborns in most African countries [44]. Similarly, in a related study in slums in Nairobi, Kenya, light porridge was given to newborns because of the belief that children from the ethnic group, *Luhya*, are always hungry right from birth [13]. Another possible explanation that can be given to this practice is that corn is one of the major staple foods in Ghana. Hence, giving corn to infants can be a way of introducing them to a common food item that they are likely to eat for the entire period of their lives. These misconceptions and cultural practices that hinder EBF practices can prevent infants from reaping the benefits of EBF such as a lowered risk of gastrointestinal infection, pneumonia, otitis media and urinary tract infections [45].

## Limitations of the study

One limitation associated with this study is that it was conducted in some communities in only two rural districts and therefore did not capture reports of other HWs from other districts in the country. However, the goal of the study was to explore misconceptions and cultural practices that acted as barriers and prevented mothers from exclusively breastfeeding their babies in this rural setting rather than generalization.

In addition, the cross-sectional study design used to collect data also makes it difficult to demonstrate cause-and-effect relationships. Regardless of these limitations, the findings provide a starting point for other surveys and interventions to reduce the impact of negative socio- cultural factors and lack of knowledge on recommended IYCF practices.

Another limitation associated with this study is that health workers were providing their viewpoints on the basis of their interactions with mothers and observations that they had made in households of community members. These viewpoints of the HWs can be described as being "second-hand" information since they were not coming directly from mothers or grandmothers. The limitation is that these perceptions of the HWs may not entirely or exactly reflect the viewpoints of mothers who were expected to practice exclusive breastfeeding. Again, there was no verification of whether HWs actually believed these socio-cultural factors perceived as influencing EBF, particularly because it could affect their delivery of health education in the community.

### Recommendations from the study

Some recommendations are made on the basis of the findings in the study. One recommendation is that, since it is difficult to change the cultural beliefs of people, there is the need for innovation and borrowing from other effective behavioural change strategies around culture in the design and implementation of interventions that target improved exclusive breastfeeding practices.

Interventions that focus on behavioural change through the giving of information, rational discussion, and skill development that can lead to the change in the attitudes and beliefs of mothers are worth considering. There is also the need for HWs to be well-informed with current recommendations on EBF in order to avoid giving contradictory information that will cast doubts in the minds of mothers regarding appropriate EBF practices.

The findings suggest the need to intensify community social mobilization programmes that target all significant stakeholders, including fathers and grandmothers, through participatory approaches, to promote best EBF practices. With regard to mothers' perceptions that HWs themselves do not practice EBF, although this was not verified, healthcare providers such as nurses and midwives should consider becoming role-models by using their own children as examples in attesting to the fact that EBF has numerous benefits to children. This role modeling may have a positive impact in preparing pregnant women or those who are still at the stage of getting breastfeeding established, and enable them gain confidence in the breastfeeding information communicated to them.

Regarding the perception of mothers that children who are exclusively breastfed do not gain weight and grow as expected, HWs could consider openly praising mothers who adhere to EBF practices; and with the consent of such mothers, use their children as examples that are worth emulating, for other mothers to also strive at improving their breastfeeding practices. Mothers and HWs who are able to exclusively breastfeed their children can also be given an opportunity to share their experiences with other mothers during CWCs.

## Conclusions

Our results show that misconceptions of mothers and grandmothers and some cultural practices as reported by the HWs continue to hinder EBF practices in the study area. The findings reveal that long-standing cultural norms that encourage mothers to give their babies water and other herbal concoctions, even immediately after birth, still persist in the study area. Generally, as reported by the HWs, grandmothers did not support EBF and they influenced the breastfeeding practices of their daughters. The revelations made by the HWs also suggests a gap in the knowledge levels of mothers regarding the adequacy of only breastmilk in satisfying the hunger and thirsty needs of babies for the first six months of life. The findings also have implications for the need to engage grandmothers, mothers-in-law and other influential community leaders like queen mothers as stakeholders in the development and implementation of interventions to promote exclusive breastfeeding practices in the districts.

## Acknowledgments

The authors highly acknowledge the data collection and management team made up of Mr. David Kwame Tsotorvor, Mrs. Victoria Parkoo, Mr Fabrics Asinyo, Mr Frank Kwasi Arthur and Mr Isaac Manford. The authors appreciate the study participants for providing a receptive environment for the study to be conducted. We also acknowledge all the support, technical advice and assistance in many ways given by the District Health Directors, Mr. Robert Kweku Bio and Mrs. Joana Amankwah, during the data collection period in the districts.

## Author Contributions

**Conceptualization:** Christiana Nsiah-Asamoah.

**Data curation:** Christiana Nsiah-Asamoah.

**Formal analysis:** Christiana Nsiah-Asamoah, David Teye Doku, Samuel Agblorti.

**Investigation:** Christiana Nsiah-Asamoah, David Teye Doku.

**Methodology:** Christiana Nsiah-Asamoah, David Teye Doku, Samuel Agblorti.

**Project administration:** Christiana Nsiah-Asamoah, David Teye Doku, Samuel Agblorti.

**Resources:** Christiana Nsiah-Asamoah, David Teye Doku, Samuel Agblorti.

**Supervision:** David Teye Doku, Samuel Agblorti.

**Validation:** David Teye Doku, Samuel Agblorti.

**Visualization:** David Teye Doku, Samuel Agblorti.

**Writing – original draft:** Christiana Nsiah-Asamoah.

**Writing – review & editing:** David Teye Doku, Samuel Agblorti.

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
