## [Decision Letter · Decision Letter 0]

15 Jun 2020

PONE-D-20-11663

Mothers' misconceptions and socio-cultural factors prevent exclusive breastfeeding: findings from two rural districts in Ghana

PLOS ONE

Dear Dr. Christiana Nsiah-Asamoah,

Thank you for submitting your manuscript to PLOS ONE. After careful consideration, we feel that it has merit but does not fully meet PLOS ONE’s publication criteria as it currently stands. Therefore, we invite you to submit a revised version of the manuscript that addresses the points raised during the review process.

We look forward to receiving your revised manuscript.

Kind regards,

Yeetey Akpe Kwesi Enuameh, MD, MSc, DrPH

Academic Editor

PLOS ONE

Journal Requirements:

2. Please include additional information regarding the interview guide used in the study and ensure that you have provided sufficient details that others could replicate the analyses. For instance, if you developed a guide as part of this study and it is not under a copyright more restrictive than CC-BY, please include a copy, in both the original language and English, as Supporting Information.In addition, please provide further details concerning participant recruitment, including the dates during which this was performed.

3.We note that you have indicated that data from this study are available upon request. PLOS only allows data to be available upon request if there are legal or ethical restrictions on sharing data publicly. For more information on unacceptable data access restrictions, please see http://journals.plos.org/plosone/s/data-availability#loc-unacceptable-data-access-restrictions.

[The authors received no specific funding for this work].

Additional Editor Comments (if provided):

The paper addresses an important issue in Maternal, Newborn and Child Health. The paper requires some revision.

1. Please address the comments of the reviewers - some very important issues have been raised by them

2. The title of the paper does not give an impression of a qualitative study... please revise

3. Reading the conclusion of the abstract creates the impression that "mothers" directly reported the issues under discussion. Please do well to project the findings as coming from healthcare workers

4. Under the "study design and population", you mention "random sampling" - that sounds like a "quantitative" approach to participant selection

5. Page 6 - the expression "breastmilk is only water and does not contain..." - was the "water" there in reference to "water" as we know it or to a "liquid"? The interpretations either way might be slightly different.

6. Page 8 - the ritual of an elder spitting into the mouth of a child - unlike the other "rituals" that seem to be common place, this one does not seem to be that documented. Did it come out overwhelmingly in the interviews or it was a one-off finding? It will be good to place it in context if it is an isolated practice. Also the health implications of such an act goes beyond breast feeding, so would need to be addressed in the discussions, conclusions and recommendations.

7. Page 12 - the last sentence of the second paragraph is not clear, please revise.

8. Also follow the guidelines for reporting qualitative studies as suggested by the reviewers

Thank you.

Reviewers' comments:

Reviewer's Responses to Questions

**Comments to the Author**

1. Is the manuscript technically sound, and do the data support the conclusions?

Reviewer #1: Yes

Reviewer #2: Yes

2. Has the statistical analysis been performed appropriately and rigorously? 

Reviewer #1: Yes

Reviewer #2: N/A

3. Have the authors made all data underlying the findings in their manuscript fully available?

Reviewer #1: No

Reviewer #2: Yes

4. Is the manuscript presented in an intelligible fashion and written in standard English?

Reviewer #1: Yes

Reviewer #2: Yes

5. Review Comments to the Author

Reviewer #1: Reviewer report

Title: Mothers' misconceptions and socio-cultural factors prevent exclusive breastfeeding:

findings from two rural districts in Ghana

Authors: Nsiah-Asamoah et al

COMMENTS

General comments

• It is a good article, highlighting misconceptions and cultural barriers and shows what the HWs know about this in relation to the mothers they deal with. It is important for HWs to understand the community beliefs so that they can appropriately deal with them, beyond only tackling the clinical interventions.

• Authors need to read through and make grammatical edits

o E.g. sentence 1 under Conclusions section in the abstract

o Sentence 1 of introduction is long and could be better edited or divided into 2, and other grammatical errors in document.

Major Compulsory Revisions

Overall methods comment

Using qualitative guidelines for paper writing will help to ensure that the methods are more fully described e.g. COREQ or other guidelines. Authors would be able to add more information on things like participant refusals or drop outs, places where interviews were held, a brief indication of what was in the FGD guides, training of research assistants, research team and reflexivity, among others.

Data analysis

• More details also needed for the analysis, for instance which kind of coding, did the coding team hold discussions to come up with themes, who did the coding, was analysis done manually or using software, was data saturation discussed, etc.

Results

Under section about grandmothers not doing EBF

• The 1st quote seems to be what HWs heard from the grandmothers, but not what the mothers themselves perceived.

• The last paragraph on that page (page 5/11) belongs to the discussion section because it is going beyond reporting results to listing their implications. The same applies to the first paragraph on page 7 (13), as well as later on the same page where authors discuss feeding of male babies, later on page 8 and in other places in the results – page 9 on herbal concoctions even cites other literature.

Limitations

• While it is good to understand the HWs view points, they are providing “second hand” information from the mothers, so it is not directly from the mothers, which is a limitation of this study.

• I think what is also missing was verification of whether HWs actually believe these things too, especially if they are from the same community. This could affect their delivery of health education.

Conclusion

• Good recommendations made. However, the difficulty in changing cultural beliefs needs to be acknowledged, and the need for innovation therein and possibly borrowing from other behavioural change interventions around culture.

Minor Essential Revisions

Introduction

• The literature and examples of barriers to EBF in the introduction only focus on cultural issues and misconceptions. There are other barriers to EBF and it would be good to briefly mention these as well.

• The last paragraph of the introduction explains why the CHWs and CHVs need to know the myths. Authors need to explain why they focused on these 2 groups only and maybe not the other HWs who deal with mothers and may be key to initiating breastfeeding at birth, for instance midwives.

• Also when authors refer to HWs, do they specifically mean only the CHWs and CHVs for this study? This needs to be clear to avoid confusion.

Study design and population

• Paragraph 1, last sentence: please specify which group of people you refer to when you say underweight. Is it children, babies, etc?

• Paragraph 2: Why should the HWs have been working for at least 5 years in the district?

• More information may be needed on the random sampling process, and how many were from CWCs and from the communities? Are these CWCs in hospitals?

• Do we have the demographics of the HWs?

• How were participants recruited? E.g. face to face, etc?

Data Collection

• Sentence 1: Focus not Focused

Data analysis

• Thematic analysis and thematic framework analysis are being used interchangeably, would you like to pick 1?

Results

• There is some repetition in the results section, for example when authors introduce the theme and then go on to the sub-themes, they need to reduce repetition of words there.

Discussion

• Remember to mention in which country or region the studies you are citing were done and possibly any limitations or strengths of the study – critique some of them.

• Highlight study strengths

Conclusion

• It may be better to have the recommendations well outlined in the discussion rather than in the conclusion.

Discretionary Revisions

Introduction

• Is there another source of breastfeeding data in Ghana beyond the GDHS? It would be a good addition to the literature, to back up the statements. This is because the DHS also has some reporting challenges. For instance, is EBF reducing or it could also be issues around reporting and data collection?

Discussion

• It would be good to elaborate on motivational theory so readers don’t have to look for it

• For studies on HWs not EBF, what were the reasons for this?

Reviewer #2: 1. Title: The title of the paper is misleading. The title suggests that the perceptions/misconceptions on exclusive breastfeeding are from the mothers’ perspective. Until you start reading, you will not have an idea that the perspectives from are health workers point of view. The authors should work on the title to reflect the views from the health workers. Example can be “Mothers misconceptions and socio-cultural factors prevent exclusive breastfeeding: Perspectives from health care providers from two rural districts of Ghana.

2. Characteristics of Health workers:

Description of CHWS and CHVs and the role they play will go a long way to help readers appreciate who these category of health providers are in the health system of Ghana. The authors will help a great deal by letting readers appreciate the context in which the study was done. E.g Who is a CHW? What does he/she do? What level of health staff is a CHW etc

3. Study design and population:

Description of the study area is very crucial to the study. It helps readers to appreciate the area where the study was conducted and also to put the results of the study in context. For instance, in the two districts where the study was done, how many health facilities are there? E.g. Is there a district hospital? Do women/community members patronise the district hospitals/health centre/CHPS compounds? The communities selected for the interviews, how close are they to the district hospital or CHPs compounds? All these will help readers appreciate the study and put the results.

Aside the health system, what is the occupation of the inhabitants in the selected communities? All these information, when provided can help to understand the people who are being studied.

4. The authors mentioned that the two districts were selected from the then Eastern Region. What is the name of the current region which host the two districts now?

5. When was the study done? How long did it take the authors from start of study to finish?

6. PLOS authors have the option to publish the peer review history of their article (what does this mean?). If published, this will include your full peer review and any attached files.

Reviewer #1: Yes: Doris Kwesiga

Reviewer #2: Yes: Charlotte Tawiah Agyemang

---

## [Author Response · Author response to Decision Letter 0]

4 Aug 2020

Many thanks for the comments and suggestions given to improve the paper, Thank you.

The table below contains the Reviewers and Academic Editor’s Comments and the authors responses. Please the line numbers stated under the authors responses refer to the revised manuscript with track changes document, thank you.

Author’s Responses to Reviewer 1

 General Comment

 Authors need to read through and make grammatical edits o E.g. sentence 1 under Conclusions section in the abstract 

 Sentence 1 of introduction is long and could be better edited or divided into 2, and other grammatical errors 

Authors Response:Please, sentence 1 under conclusions section of the abstract has been corrected, please refer to lines 34 and 35.

Please, sentence 1 of the introduction has been divided into two sentences, please see lines 46 - 49. 

Overall methods comment

Using qualitative guidelines for paper writing will help to ensure that, the methods are fully described. E.g COREQ or other guidelines 

Authors Response:Many thanks and well appreciated. We went through the COREQ guidelines and the manuscript has been revised accordingly to a large extent.

 Authors would be able to add more information on things like participant refusals or drop outs, places where interviews were held, a brief indication of what was in the FGD guides, training of research assistants, research team and reflexivity, among others 

Authors Response:Thank you for the comments. Please, in this study there were no issues with participant refusals or drop outs.

- The places where the FGDs were conducted, that is, at the District Health Directorate’s office has been indicated in the revised manuscript, please see lines 191 and 192, under Data collection section.

- A brief indication of what was in the FGD guides has been added to the revised manuscript, please see lines 196 - 205.

- A short section on the research team and reflexivity have been added to the revised manuscript, please refer to lines 239 - 247.

Data Analysis

More details also needed for the analysis, for instance which kind of coding, did the coding team hold discussions to come up with themes, who did the coding, was analysis done manually or using software, was data saturation discussed, etc. 

Authors Response:Many thanks. The data analysis section has been revised to address the comments raised, please see lines 219 - 237. 

- Regarding discussion of data saturation by the research team, please refer to lines 209 - 212.

Results

Under section about grandmothers not doing EBF 

• The 1st quote seems to be what HWs heard from the grandmothers, but not what the mothers themselves perceived.

Authors Response:Thank you for the observation. We agree some of the quotes were what HWs had heard from grandmothers and not mothers. Please, the title has been revised to include grandmothers, please refer to lines 3 -5. 

Results

-The last paragraph on that page (page 5/11) belongs to the discussion section because it is going beyond reporting results to listing their implications. 

-The same applies to the first paragraph on page 7 (13), as well as later on the same page where authors discuss feeding of male babies, later on page 8 and in other places in the results – page 9 on herbal concoctions even cites other literature 

Limitations

-While it is good to understand the HWs view points, they are providing “second hand” information from the mothers, so it is not directly from the mothers, which is a limitation of this study. 

-I think what is also missing was verification of whether HWs actually believe these things too, especially if they are from the same community. This could affect their delivery of health education. 

Authors Response:Thank you for the observation. The various sections on pages 5-11 which discuss the findings have been deleted from the results section and moved to the discussion section as shown in track changes on lines 303-306, 327-331, 378-382, 420-423, 463- 467.

Limitations

Authors Response:Thank you for the comment. We agree the HWs are providing “second hand” information based on their encounters with mothers, grandmothers and observations made in the community. This limitation of the study has been captured under the limitation section, please see lines 664 - 670.

Authors Response:Please, we agree that this is also another limitation of the study and has been indicated under the limitation section, please refer to lines 670-672.

Conclusion

-Good recommendations made. 

-However, the difficulty in changing cultural beliefs needs to be acknowledged, and the need for innovation therein and possibly borrowing from other behavioural change interventions around culture. 

Authors Response:Thank you, please, a sentence has been inserted under the recommendations from study section acknowledging the difficulty associated with changing cultural beliefs of people and as such the need for innovation and borrowing from other effective behavioural change interventions around culture. Please refer to lines 675 - 681.

Minor Essential Revisions 

Introduction 

-The literature and examples of barriers to EBF in the introduction only focus on cultural issues and misconceptions. There are other barriers to EBF and it would be good to briefly mention these as well. 

Authors Response: Many thanks for the suggestion. Please, in the introduction section, other barriers to EBF have been indicated briefly, please refer to lines 70 -78.

Introduction

-The last paragraph of the introduction explains why the CHWs and CHVs need to know the myths. Authors need to explain why they focused on these 2 groups only and maybe not the other HWs who deal with mothers and may be key to initiating breastfeeding at birth, for instance midwives. 

Authors Response:In the last paragraph of the introduction, an explanation has been given on why the study focused on CHWs and CHVs and not the other categories of health workers.

Please refer to lines 102 -116.

Introduction

- Also, when authors refer to HWs, do they specifically mean only the CHWs and CHVs for this study? This needs to be clear to avoid confusion. 

Authors Response:

Authors Response: Many thanks. Please, the abbreviation HWs refers collectively to the study participants in this study (CHWs and CHVs). An insertion has been made in the last sentence of the introduction -(collectively referred to as HWs for the purpose of this study) to indicate this. Please see lines 125 - 127.

Study design and population 

-Paragraph 1, last sentence: please specify which group of people you refer to when you say underweight. Is it children, babies, etc? 

Authors Response:Please, underweight refers to children under five years of age in the region. This insertion has been made, please refer to line 174.

-Paragraph 2: Why should the HWs have been working for at least 5 years in the district? 

Authors Response:This was an assumption that was made by us (the authors). The assumption was that this number of years might have given HWs ample experiences to be able to provide information on socio-cultural and misconception issues influencing EBF within the selected districts. This information has been added to the section study design and population. Please see lines 179 -182

-More information may be needed on the random sampling process, and how many were from CWCs and from the communities? Are these CWCs in hospitals? 

Authors Response:The random sampling refers to the selection of the CWCs in various health facilities within the districts. No please, these CWCs are not units within only hospitals but the various health facilities such as health centres, CHPS compounds etc. Please refer to lines 175- 186. 

-Do we have the demographics of the HWs.

Authors Response:Please, the only demographic information we have of the HWs are the sex, work experience(in years) and whether they have participated in any workshop on child nutrition issues and a table has now been presented in the revised manuscript. Please refer to Table 1, lines 255 -258

-How were participants recruited? 

E.g. face to face, etc? 

Authors Response:Please, the participants were recruited face-to-face with assistance from the district nutrition officers and nurse-in-charges in the various CWCs. This information has been added to lines 182 - 186.

Data Collection

-Sentence 1 Focus not Focused.

Authors Response:Thank you. Please “focus” has been changed to “focused", refer to line 193.

Data analysis 

-Thematic analysis and thematic framework analysis are being used interchangeably, would you like to pick 1? 

Authors Response:Thank you for the observation, we will use Thematic analysis throughout the manuscript.

Results 

-There is some repetition in the results section, for example when authors introduce the theme and then go on to the sub-themes, they need to reduce repetition of words there. 

Authors Response:Well appreciated. Please, we have read through the results section again and all repetitions in the main theme and sub-themes have been deleted. Please see lines 290, 346, 

Discussion

-Remember to mention in which country or region the studies you are citing were done and possibly any limitations or strengths of the study – critique some of them. 

Authors Response:Thank you for the comment. All the countries in which the studies that have been cited in the discussion were conducted have been indicated. Please refer to the discussion section on lines 508, 516, 582, 583 and 645.

Conclusion 

-It may be better to have the recommendations well outlined in the discussion rather than in the conclusion. 

Authors Response:Thank you for the suggestion. The recommendations have been moved to the last paragraph of the discussion, please refer to lines 674 - 697.

Discretionary Revisions 

Introduction 

-Is there another source of breastfeeding data in Ghana beyond the GDHS? It would be a good addition to the literature, to back up the statements. This is because the DHS also has some reporting challenges. For instance, is EBF reducing or it could also be issues around reporting and data collection. 

Authors Response:Many thanks. We agree that the GDHS has limitations such as reporting challenges. However, it is only the GDHS that gives a holistic and comprehensive information regarding child nutrition in Ghana. For instance, World Bank and UNICEF reports on the state of child nutrition in Ghana make reference to the GDHS data and findings. 

Discussion 

- It would be good to elaborate on motivational theory so readers don’t have to look for it.

- For studies on HWs not practicing EBF, what were the reasons for this? 

Authors Response:Thank you for the suggestion, Please, more information about the motivational theory has been indicated in lines 528 -535. 

Authors Response:Some reasons for not practicing EBF by nurses and midwives in studies that were cited in the discussion have been inserted in lines 517-520.

Author’s Responses to Reviewer 2

Reviewer’s Comment Author’s response

1. Title: The title of the paper is misleading. The title suggests that the perceptions/misconceptions on exclusive breastfeeding are from the mothers’ perspective. Until you start reading, you will not have an idea that the perspectives from are health workers point of view. The authors should work on the title to reflect the views from the health workers. Example can be “Mothers misconceptions and socio-cultural factors prevent exclusive breastfeeding: Perspectives from health care providers from two rural districts of Ghana. 

Authors Response:Thank you for the suggestion. Please, we have revised the title to reflect the content of the manuscript.

2. Characteristics of Health workers:

Description of CHWS and CHVs and the role they play will go a long way to help readers appreciate who these category of health providers are in the health system of Ghana. The authors will help a great deal by letting readers appreciate the context in which the study was done. E.g Who is a CHW? What does he/she do? What level of health staff is a CHW etc 

Authors Response:Many thanks for the comments. Please, some information on the CHWs and CHVs has been provided in the introduction. Please refer to lines 102-116.

3. Study design and population:

Description of the study area is very crucial to the study. It helps readers to appreciate the area where the study was conducted and also to put the results of the study in context. For instance, in the two districts where the study was done, how many health facilities are there? E.g. Is there a district hospital? Do women/community members patronise the district hospitals/health centre/CHPS compounds? The communities selected for the interviews, how close are they to the district hospital or CHPs compounds? All these will help readers appreciate the study and put the results.

Authors Response:Thank you for the comment. A sub-heading “Study Area” has been included under the “Subjects and Methods” section which describes the study area. Please refer to lines 141-166.

Aside the health system, what is the occupation of the inhabitants in the selected communities? All this information, when provided can help to understand the people who are being studied. Authors Response:Thank you for the suggestion. Information regarding the occupation of the inhabitants has been provided under the section “Study Area”, please see lines 148-150, 158-159.

4. The authors mentioned that the two districts were selected from the then Eastern Region. What is the name of the current region which host the two districts now? 

Authors Response:Thank you for the observation. This is an oversight, the name of region remains the same -Eastern Region. The word “then has been deleted from the 1st and 2nd sentences under the section “Study Design and Population” Please see lines 170, 172.

5. When was the study done? How long did it take the authors from start of study to finish? Authors Response:Please, information about the period and duration of the study has been provided under the Data collection section. Please refer to lines 189 - 192.

 Responses to Academic Editor’s Comments

Academic Editor’s Comments Author’s Comments

1.The title of the paper does not give an impression of a qualitative study... please revise Authors Response:Thank you for the suggestion. Please, the title has been revised to show that it is a qualitative study.

Reading the conclusion of the abstract creates the impression that "mothers" directly reported the issues under discussion. Please do well to project the findings as coming from healthcare workers. 

Authors Response:Well appreciated. The beginning sentence of the conclusion section under the Abstract has been revised to indicate that the issues discussed in the study are from health workers. Please refer to lines 34-35.

Under the "study design and population", you mention "random sampling" - that sounds like a "quantitative" approach to participant selection.

Authors Response:Please, the random sampling applies to the selection of the 21 Child Welfare Clinics (CWCs). Please it does not refer to the random selection of the participants(CHWs).

 From each CWC, the nurse-in-charge assisted in selecting 2 CHWs to participate in the FGD. This has been presented under the study “design and population” section, lines 175 -177.

Page 6 - the expression "breastmilk is only water and does not contain..." - was the "water" there in reference to "water" as we know it or to a "liquid"? The interpretations either way might be slightly different. 

Authors Response:Many thanks for the observation. Please, this statement has been revised. The HWs meant some mothers have the perceptions that, breastmilk is more watery in nature and does not contain enough food to satisfy the hunger needs of the child. Please refer to lines 366, 368 and 370.

Page 8 - the ritual of an elder spitting into the mouth of a child - unlike the other "rituals" that seem to be common place, this one does not seem to be that documented. Did it come out overwhelmingly in the interviews or it was a one-off finding? It will be good to place it in context if it is an isolated practice. Also the health implications of such an act goes beyond breast feeding, so would need to be addressed in the discussions, conclusions and recommendations. 

Authors Response:The comment is well appreciated. This report is a one-off finding that was reported in one community which has several Fulani herdsmen households. The authors have therefore deleted it from the manuscript since the practice did not come out overwhelmingly in all the 9 FGDs that were conducted. Please see the deletion in lines 451 - 454.

Page 12 - the last sentence of the second paragraph is not clear, please revise. 

Authors Response:Well appreciated. The sentence was making reference to the finding on the ritual of an elder person spitting into the mouth of a newborn baby. The sentence has been deleted since this finding which did not come out overwhelmingly in the FGDs has also been deleted from the results section. 

Also follow the guidelines for reporting qualitative studies as suggested by the reviewers.

Authors Response: Many thanks and well appreciated. Please, we went through the COREQ guidelines and the manuscript has been revised accordingly to a large extent.

---

## [Decision Letter · Decision Letter 1]

3 Sep 2020

Mothers’ and Grandmothers’ misconceptions and socio-cultural factors as barriers to exclusive breastfeeding: A qualitative study involving Health Workers in two rural districts of Ghana

PONE-D-20-11663R1

Dear Dr. Christiana Nsiah-Asamoah,

We’re pleased to inform you that your manuscript has been judged scientifically suitable for publication and will be formally accepted for publication once it meets all outstanding technical requirements.

Kind regards,

Yeetey Akpe Kwesi Enuameh, MD, MSc, DrPH

Academic Editor

PLOS ONE

Additional Editor Comments (optional):

Great effort on the part of the authors... Most of the issues raised have been addressed satisfactorily. Though the manuscript has been passed for acceptance, the authors should address the issues raised by reviewer 1. As mentioned there are a few grammatical and some structural errors. The authors seem to love "long" sentences - some sentences in the changes are still overly long making appreciation of the concepts not so easy. Fragmenting and rewording these would help bring issues out clearly. Under the methods, FGD was spelt as FDG... this and a few others should be corrected. Please finally read over carefully and possibly get a proof reader to go over to ensure any errors are picked up and addressed.

Thank you.

Reviewers' comments:

Reviewer's Responses to Questions

**Comments to the Author**

1. If the authors have adequately addressed your comments raised in a previous round of review and you feel that this manuscript is now acceptable for publication, you may indicate that here to bypass the “Comments to the Author” section, enter your conflict of interest statement in the “Confidential to Editor” section, and submit your "Accept" recommendation.

Reviewer #1: (No Response)

Reviewer #2: All comments have been addressed

2. Is the manuscript technically sound, and do the data support the conclusions?

Reviewer #1: Yes

Reviewer #2: (No Response)

3. Has the statistical analysis been performed appropriately and rigorously? 

Reviewer #1: Yes

Reviewer #2: (No Response)

4. Have the authors made all data underlying the findings in their manuscript fully available?

Reviewer #1: Yes

Reviewer #2: (No Response)

5. Is the manuscript presented in an intelligible fashion and written in standard English?

Reviewer #1: Yes

Reviewer #2: (No Response)

6. Review Comments to the Author

Reviewer #1: General comments

• Noted the change in title to include grandmothers as well. The option suggested by the other reviewer was also good.

• Most of the changes have been made as recommended in the first review, and the manuscript has been greatly improved. I only recommend a few minor revisions as below.

• In the abstract it is now explained that you are looking at the perspectives of health workers so that is clearer.

• Authors need to read through and make grammatical edits, especially shortening the very long sentences, which can be broken into two. For instance, these are still present in the abstract, although authors worked on some. There are also areas without good punctuation. You can also reduce repetitions on page 14 – the section explaining why focus of this study was on CHVs and CHWs. Focused group discussions still mentioned on page 16 under data collection. Another is the title for the first theme in the results, and many others. The discussion and conclusion sections would be easier to read with more paragraphs than the big chunks of text.

Minor Essential Revisions

Introduction

If GDHS is the only data source please include a sentence specifying the limitations of this, considering it’s 2014 data

Methods comment

It would be good to indicate whether the lead researchers and the team were Ghanaian and local to the area or not.

Discussion

• Remember to briefly critique some of the other studies cited

• The discussion now has the right content. Authors can focus on editing it to make it more concise, with less repetition of results and more focus on contextualising their study

Conclusion

• It mentions a gap in mothers knowledge but I think grandmothers should be included as well.

Reviewer #2: (No Response)

7. PLOS authors have the option to publish the peer review history of their article (what does this mean?). If published, this will include your full peer review and any attached files.

Reviewer #1: **Yes: **Doris Kwesiga

Reviewer #2: **Yes: **Charlotte Tawiah Agyemang

---

## [Editor Report · Acceptance letter]

8 Sep 2020

PONE-D-20-11663R1 

Mothers’ and Grandmothers’ misconceptions and socio-cultural factors as barriers to exclusive breastfeeding: A qualitative study involving Health Workers in two rural districts of Ghana 

Dear Dr. Nsiah-Asamoah:

I'm pleased to inform you that your manuscript has been deemed suitable for publication in PLOS ONE. Congratulations! Your manuscript is now with our production department. 

Kind regards, 

on behalf of

Dr. Yeetey Akpe Kwesi Enuameh 

Academic Editor

PLOS ONE